# Common Sea Star (*Asterias rubens*) Coelomic Fluid Changes in Response to Short-Term Exposure to Environmental Stressors

Sarah J. Wahltinez [1], Kevin J. Kroll [2], Donald C. Behringer [3,4], Jill E. Arnold [5], Brent Whitaker [5], Alisa L. Newton [5,6], Kristina Edmiston [1], Ian Hewson [7] and Nicole I. Stacy [1,*]

1    Department of Comparative, Diagnostic, and Population Medicine, College of Veterinary Medicine, University of Florida, Gainesville, FL 32610, USA
2    Department of Physiological Sciences, Center for Environmental and Human Toxicology, College of Veterinary Medicine, University of Florida, Gainesville, FL 32610, USA
3    School of Forest, Fisheries, and Geomatics Sciences, University of Florida, Gainesville, FL 32611, USA
4    Emerging Pathogens Institute, University of Florida, Gainesville, FL 32610, USA
5    ZooQuatic Laboratory LLC, Baltimore, MD 21202, USA
6    OCEARCH, Park City, UT 84068, USA
7    Department of Microbiology, Cornell University, Ithaca, NY 14853, USA
*    Correspondence: stacyn@ufl.edu

**Abstract:** Common sea stars (*Asterias rubens*) are at risk of physiological stress and decline with projected shifts in oceanic conditions. This study assessed changes in coelomic fluid (CF) blood gases, electrolytes, osmolality, and coelomocyte counts in adult common sea stars after exposure to stressors mimicking effects from climate change for 14 days, including decreased pH ($-0.4$ units, mean: 7.37), hypoxia (target dissolved oxygen ~1.75 mg $O_2$/L, mean: 1.80 mg $O_2$/L), or increased temperature ($+10$ °C, mean: 17.2 °C) and compared sea star CF electrolytes and osmolality to tank water. Changes in CF blood gases, electrolytes, and/or coelomocyte counts occurred in all treatment groups after stressor exposures, indicating adverse systemic effects with evidence of increased energy expenditure, respiratory or metabolic derangements, and immunosuppression or inflammation. At baseline, CF potassium and osmolality of all groups combined were significantly higher than tank water, and, after exposures, CF potassium was significantly higher in the hypoxia group as compared to tank water. These findings indicate physiological challenges for *A. rubens* after stressor exposures and, given increased observations of sea star wasting events globally, this provides evidence that sea stars as a broad group are particularly vulnerable to changing oceans.

**Keywords:** blood gases; coelomocyte counts; electrolytes; osmolality; point-of-care analyzer; starfish





## 1. Introduction

Common sea stars (*Asterias rubens*) are medium sized, typically 10–30 cm in diameter from ray tip to ray tip [1], and are found extensively throughout the eastern and western North Atlantic Ocean from the intertidal zone to 650 m depth [2,3]. They are considered a keystone species of the North Atlantic rocky intertidal zone because they control mussel populations, which provides space for other species, thereby increasing ecosystem biodiversity [2–6]. *Asterias* is one of many genera affected by sea star wasting (SSW) [7], which is a complex disorder of sea stars exhibiting a suite of clinical signs that include epidermal lesions followed by autotomy, body wall disintegration, and often death. Sea star wasting has been observed in a variety of sea star species worldwide [8], but a definitive cause has not yet been identified for any recent sea star mass mortality event. Reports of these events have increased in frequency over the last decade, and the number of affected species, individuals, and geographical ranges point towards environmental changes to sea star habitat as a significant factor [8,9]. Oceanic changes that may impact sea stars include stressors associated with climate change, including acidification, hypoxia (decreased dissolved oxygen), and warming.

Ocean acidification is the decrease in ocean pH driven by air–water gas exchange of gases, primarily carbon dioxide, from the atmosphere into oceans [10]. Atmospheric $pCO_2$ concentrations have risen nearly 40% since the Industrial Revolution, primarily driven by the combustion of fossil fuels from human activity [10]. When carbon dioxide enters the ocean, it reacts with seawater to form carbonic acid, a weak acid that quickly disassociates to form bicarbonate and free hydrogen ions. Free hydrogen ions decrease the pH and also react with carbonate to form additional bicarbonate. This shifts the equilibrium from carbonate towards carbonic acid and as a result, the pH of the ocean decreases: $CO_{2(Atoms)} \rightleftharpoons CO_{2(Aqueous)} + H_2O \rightleftharpoons H_2CO_3 \rightleftharpoons H^+ + HCO_3^- \rightleftharpoons 2H^+ + CO_3^{2-}$ [10]. The current ocean pH is approximately 8.1 [10]. The Intergovernmental Panel on Climate Change (IPCC) concluded in 2007 that the decrease in ocean water pH could reach 0.3–0.4 units by the year 2100 and 0.8 by the year 2300 under the "business-as-usual" IS292a scenario [11]. Common sea stars appear to be limited in their ability to mount an adaptive physiological response and are unable to compensate their coelomic fluid pH when exposed to seawater at pH 7.4 and 7.7 [12]. In another study, exposure to seawater at pH 7.7 resulted in immunosuppressive effects on common sea stars, which exhibited a 50% decrease in circulating coelomocytes, reduced coelomic fluid pH, and increased levels of 70 kDa heat shock protein (Hsp70) [13]. Heat shock proteins are chaperone proteins that assist in the protein folding process and often increase in response to stressors [14].

In aquatic ecosystems, hypoxia is commonly defined as dissolved oxygen (DO) concentrations of <2.0 mg $O_2$/L (or ~20% oxygen saturation at 13 °C and 32 ppt salinity) [15,16]. In coastal ecosystems globally, the occurrence of hypoxic conditions has expanded at a rate of 5.54% per year, resulting from eutrophication, organic matter enrichment, increased temperatures, or a combination of these factors [16]. Sea stars respire using passive diffusion through specialized gills called papulae, which are evaginations of coelomic epithelium on the aboral surface, and tube feet [17,18]. Due to limited compensatory mechanisms, sea stars may be more sensitive to decreased DO availability. However, potentially due to low metabolic rates, the median sublethal oxygen concentration for echinoderms has been determined to be $1.22 \pm 0.22$ mg $O_2$/L [16]. Lined sea stars (*Luidia clathrata*) have been reported to tolerate DO levels of 2.5 mg $O_2$/L for short periods of time [19]. In common sea stars, 50% mortality was observed at 84 h in 0.2 mg $O_2$/L [20]. Exposure to hypoxia has been associated with increased coelomocyte counts and increased levels of Hsp70 [21]. Oxygen limitation may also occur as a result of shifting body wall microbial communities due to organic matter enrichment, leading to a diffusive boundary layer impeding oxygen exchange. In laboratory exposures, organic matter enrichment and the resultant hypoxia were associated with lesions similar to those seen in SSW [22].

Ocean warming occurs as oceans act as a heat sink for the extra heat accumulating on Earth due to the greenhouse effect [23]. Ocean temperatures are projected to increase an average of 0.5 °C per decade [24]. While much attention has been given to increased risk of infectious disease [25], there are other consequences of increasing temperature that may lead to stress or mortality, such as increased metabolism [26,27] in the face of decreased feeding rates [28,29] and induced production of heat shock proteins, which requires further energy input [28]. Increased water temperatures have been associated with mortality in multiple species of sea stars. A temperature increase of 8–13 °C was associated with a mortality event in the burrowing sea star *Astropecten jonstoni* [30]; a greater proportion of ochre sea stars (*Pisaster ochraceus*) had evidence of SSW and an increased severity with a 4 °C temperature increase [31]; and a 4 °C surface water temperature increase was associated with increased clinical evidence of SSW in the sunflower star (*Pycnopodia helianthoides*) [32].

Analysis of coelomic fluid and its coelomocytes can be used to evaluate the health status, immune or inflammatory responses, and physiological changes associated with various disease states, which may provide insight into pathophysiological mechanisms. Coelomic fluid is analogous to blood in vertebrate animals and bathes the internal organs of sea stars [17]. Coelomocytes, the circulating cell type in coelomic fluid, are the effector

cells of the echinoderm immune system [33,34]. Coelomic fluid is an informative sample matrix due to its close proximity to coelomic organs and non-lethal accessibility.

The goal of this study was to determine the effects of environmental stressors on adult common sea stars by assessing coelomic fluid changes as indicators of physiological stress. This study evaluated changes in coelomic fluid blood gases, electrolytes, osmolality, and coelomocyte counts after exposure to stressors mimicking effects of climate change for 14 days, including decreased pH ($-0.4$ units), hypoxia (target DO ~1.75 mg $O_2$/L), or increased temperature (+10 °C), and compared sea star coelomic fluid electrolytes and osmolality to tank water.

## 2. Materials and Methods

### 2.1. Sea Star Collection and Husbandry

Ethical and animal welfare principles of veterinary medicine were followed during this research study. Sea stars were handled and housed in accordance with guidelines for aquatic species published in the Guide for the Care and Use of Laboratory Animals, 8th edition [35].

Free-ranging adult common sea stars (*Asterias rubens*) were collected sub-tidally from the same population used by commercial collectors (Ocean Resources Inc., Sedgwick, ME, USA) in May 2021. Each sea star was placed in a perforated plastic bag, with two sea stars per larger plastic bag, with a small amount of water and air. Sea stars were packed separately from icepacks and shipped overnight to the University of Florida. Upon arrival, sea stars were acclimated over 6 hours to new tank conditions.

Sea stars were housed with a 12:12 (light:dark) photoperiod, air temperature ranging from 15.6–22.8 °C, and humidity from 20–75%. Prior to enrollment, sea stars were group housed in natural Atlantic Ocean seawater in six 380 L fiberglass tanks with a heat exchanger, air stone, mechanical filtration with a canister filter, and biological filtration with a fluidized sand bed filter. All biological filters were seeded with live nitrifying bacteria (Fritzyme® TurboStart 900 Saltwater®, Fritz Aquatics, Mesquite, TX, USA). Sea stars were fed one frozen-thawed clam or mussel once weekly (Hikari®, Kyorin Food Industries, Ltd., Hyogo, Japan; Mi, Newark, CA, USA; Pro Salt, Mid Jersey Pet Supply, Carteret, NJ, USA; V2O Aquarium Foods, LLC, Layton, UT, USA). Water quality was tested daily on a rotating basis, with each tank tested once every 5 days for ammonia ($NH_3$-N), nitrite ($NO_2$-N), and nitrate ($NO_3$-N) with a colorimeter (Hach DR 900 Portable Colorimeter, Hach Company, Loveland, CO, USA); alkalinity by drip test kit (Salifert Worldwide, Duiven, Holland); and salinity, pH, DO and temperature with a multiparameter water quality meter (ProDSS Digital, YSI Inc., Yellow Springs, OH, USA). The NBS scale was used for all tank water pH measurements. Salinity was calibrated monthly, pH was calibrated weekly using 3 standard solutions (pH 4.0, 7.0, 10.0), and DO was calibrated daily, per manufacturer instructions. Solid waste was removed daily or more frequently as needed. Water changes were performed as indicated by increased ammonia (>0.2 mg/L), nitrite (>0.05 mg/L) or nitrate (>5 mg/L). A maximum 50% volume replacement per change was performed during exposures.

Sea star behavior was observed once daily during exposures and recorded as normal or abnormal (e.g., twisted arms, inflated appearance, visible areas of epidermal ulceration, arm autotomy, etc.). Sea stars that appeared grossly abnormal were photographed and monitored closely for progression of clinical signs that would warrant euthanasia.

### 2.2. Coelomic Fluid Sampling Procedure

Sea stars were group housed for at least 4 months prior to enrollment in this study. While group housed, sea stars appeared to feed and behave normally. For an overview of exposures and sampling, see Figure 1.

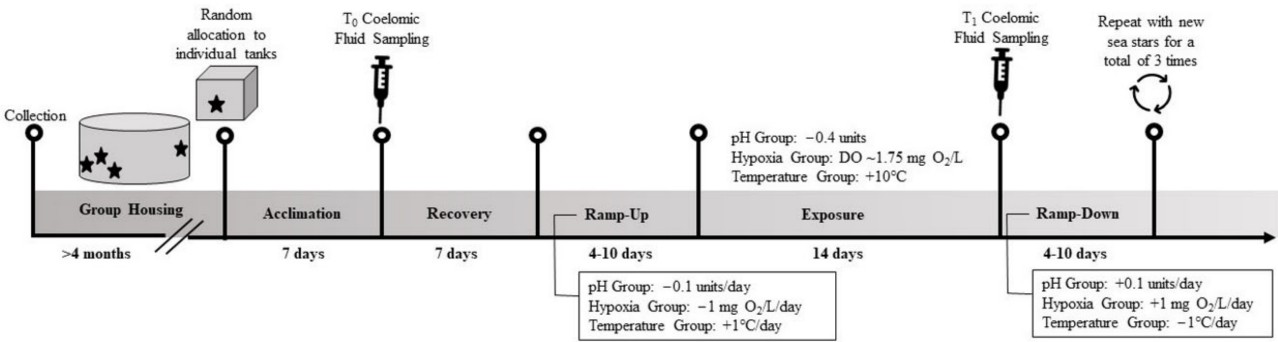

**Figure 1.** Visual overview of environmental stressor exposure for adult common sea stars (*Asterias rubens*).

Sea stars were randomly selected from group housing tanks and moved to individual 53 L tanks equipped with a heat exchanger, air stone, mechanical sock filter and fiber filtration media (Matala Water Technology, Taichung City, Taiwan) for biological filtration. Water quality was tested as above, with pH, DO, and temperature recorded for each tank daily. Sea stars were assigned to one of four groups: pH, hypoxia, temperature, and control (*n* = 5 per group, repeated three times for *n* = 15 replicates per treatment). Identification numbers, treatment group and tank number were randomly generated. Wet weight (g) and contoured diameter (cm, the distance from the visually longest tip of one ray across the central disc to the tip of the ray directly across) were recorded as sea stars were moved to individual tanks. Sea stars were allowed to acclimate in individual tanks for 7 days before baseline coelomic fluid samples were taken. Inclusion criteria for enrollment were feeding during the acclimation period, no visible external lesions, and firm attachment to tank walls.

On the day of baseline sampling ($T_0$), sea stars were weighed, then returned to their tanks for a minimum of 15 min. The maximum amount of coelomic fluid removed per 24 h period was 2% of total wet body weight. Assuming coelomic fluid volume is 20% of the sea star's total wet body weight [36], this is similar to current recommendations for safe withdrawal of up to 10% of the total blood volume in small vertebrate animals [37,38]. An initial coelomic fluid sample was collected for immediate blood gas analysis. Sea stars were held above their tank, and 0.1 mL coelomic fluid was immediately (within 10 s) collected from the perivisceral coelomic cavity using a 23-gauge 2.5 cm needle (Monoject, Covidien, Minneapolis, MN, USA) and 1 mL tuberculin syringe (VetriJec, VetOne, Boise, ID, USA) approximately 1 cm from the distal tip of a ray on the aboral surface of the sea star [39]. The sea star was returned to the tank, and coelomic fluid was immediately (within 10 s) analyzed using a point-of-care blood gas analyzer (iSTAT CG4+ cartridge, Abbott Diagnostics, Lake Forest, IL, USA). Coelomic fluid analytes measured included pH, partial pressure of carbon dioxide ($pCO_2$), partial pressure of oxygen ($pO_2$), and lactate, while bicarbonate ($HCO_3$) and total carbon dioxide ($TCO_2$) were calculated. Since the blood gas analyzer warms samples to 37 °C prior to analysis, a temperature correction using the average tank temperature was applied to pH, $pCO_2$ and $pO_2$ prior to statistical analysis to obtain $pH_{(TC)}$, $pCO_{2(TC)}$, and $pO_{2(TC)}$ [40]. While the point-of-care blood gas analyzer produced results for base excess and oxygen saturation ($sO_2$), these data were excluded from this study [41].

After blood gas analyses were complete, sea stars were placed individually in a transport container with 1 L of its tank water, and sampling was continued. Another coelomic fluid sample of the remaining safe-removal volume (e.g., 2% of total wet body weight) was collected as above; 600 μL were placed in a sterile cryogenic vial with no additives for analysis of magnesium (Mg), sodium (Na), potassium (K), chloride (Cl), calcium (Ca), and osmolality and frozen immediately in liquid nitrogen, then stored at −80 °C prior to analysis; 200 μL were saved in a sterile microcentrifuge tube for cytological evaluation and stored on wet ice, and 50 μL were pipetted into 200 μL 10% neutral buffered

formalin [42] and stored at room temperature for coelomocyte counts. Contoured diameter (cm) was measured, and the sea star was photographed prior to being returned to its tank. At the time of coelomic fluid sampling, 50 mL water samples were collected and frozen at $-80\,°C$ prior to analysis.

Within 3 h from time of collection, coelomic fluid was prepared for cytological evaluation using a cytocentrifuge (Cytopro$^{TM}$, Wescor, South Logan, UT, USA). Slides were stained with Wright Giemsa stain (Harleco$^{®}$, EMD Millipore, Billerica, MA, USA) prior to review by one investigator (N.I.S.) who was blinded to sea star treatment groups. Manual coelomocyte counts were performed on formalin-fixed preparations as has been previously described [42], with consideration of the dilution factor resulting from the formalin dilution. Coelomic fluid and water samples were analyzed for Mg, Na, K, Cl, Ca, and osmolality using the same methods within 6 months of collection. Diluted samples (1:50) were analyzed by spectrophotometry for Mg (ChemWell-T, CATACHEM, INC., Oxford, CT, USA). Diluted samples (1:3) were analyzed by ion selective electrode for Na, K, and Cl (Axcel, Alfa Wasserman, West Caldwell, NJ, USA) and by spectrophotometry for Ca (Axcel, Alfa Wasserman, West Caldwell, NJ, USA). The coefficient of variation for electrolyte analyses ranged from 2.0–2.4% (Table S1). Osmolality was measured on undiluted samples using a vapor pressure osmometer (Wescor Inc., Logan, UT, USA) after calibration using osmolality standards 290 mmol/Kg and 1000 mmol/Kg, and the 1000 mmol/Kg standard was repeated after every 10 samples to confirm there was no drift in calibration (ELITechGroupTM, Wescor Inc., Logan, UT, USA.

Following $T_0$, sea stars were given an additional 7 days to recover before a gradual ramp-up to target environmental stressor levels was initiated. These target levels were then maintained for 14 days. After 14 days of exposure, coelomic fluid sampling was repeated ($T_1$) as for baseline sampling.

Following $T_1$, a gradual ramp-down of stressor levels was performed, the reverse of the ramp-up period, out of concern for animal welfare and to allow sea stars to adjust to their baseline housing parameters. Sea stars were then returned to group housing and another group of sea stars were enrolled. Between repetitions, tanks were completely drained, rinsed with natural salt water, and refilled. Mechanical filters were pressure washed with freshwater. Tanks were not sterilized to preserve biological filtration; concerns over pathogen transmission were limited as these sea stars were from the same source population and had been group housed prior to enrollment.

*2.3. Experimental Exposures*

For the pH group, tank pH was decreased using a pH controller (MC122 PRO, Milwaukee Instruments, Inc., Rocky Mount, NC, USA) and Pure Clean grade $CO_2$ gas (Air Gas, Radnor, PA, USA) bubbled through a 5.1 cm fine-pore ceramic air stone (Pentair Aquatic Eco-Systems, Inc., Cary, NC, USA). During the ramp-up period, pH was decreased 0.1 units per day to a target of a 0.4-unit decrease. This is consistent with the IPCC projections of a decrease in mean ocean pH of 0.3–0.4 units by the year 2100 [11]. In accordance with the Guide to Best Practices for Ocean $CO_2$ Measurements [43], water samples were taken from three randomly selected tanks in the pH and control groups for total dissolved inorganic carbon (DIC) and total alkalinity at day 0, day 4 (end of ramp-up), day 11 (7 days of exposure) and day 18 (14 days of exposure). For DIC analyses, water samples were collected using a 12 mL syringe and filtered through a 0.22 μm nylon syringe filter (Thermo Scientific, Waltham, MA, USA) into a 7 mL glass screw cap vial with no headspace. Samples were refrigerated until analysis. Total DIC was measured using an automated acidification unit (AutoMate Prep Device, AutoMate FX, Inc., Bushnell, FL, USA) and a carbon dioxide coulometer (CM5017, UIC, Inc., Joliet, IL, USA). For total alkalinity, 50 mL water samples were collected in a screw top conical tube and frozen at $-80\,°C$ until analysis. Total alkalinity was measured by titration, using Standard Methods 2320 [44] and a flow injection analyzer (Dionex, ThermoFisher Scientific, Waltham, MA, USA) after daily

calibration by standard curve using anion and cation standards (SPEXCertiPrep, Metuchen, NJ, USA).

For the hypoxia group, DO was decreased using Ultra High Purity grade $N_2$ gas (Air Gas, Radnor, PA, USA) bubbled through micropore diffusers (Point Four™ Plastic Micro Bubble Diffuser, Pentair Aquatic Eco-Systems, Inc., Cary, NC, USA). The desired levels of DO were achieved by manually adjusting a hose clamp with screw (Gilson Company, Inc., Lewis Center, OH, USA) on a pressurized gas line suppling $N_2$ gas to each tank. During the ramp-up period, DO was decreased ~1 mg $O_2$/L per day to a target of 1.75 mg $O_2$/L. Hypoxia is defined as $\leq 2$ mg $O_2$/L [15,16], and that cut-off is used by the Environmental Protection Agency and United States Geological Survey.

For the temperature group, water temperature was increased by adjusting the heat exchanger (AquaLogic Inc., San Diego, CA, USA) on each tank. During the ramp-up period, temperature was increased 1 °C per day to a target increase of 10 °C above ambient housing temperature (17 °C), simulating increased ocean temperatures. NOAA buoy I01 in the eastern Maine Shelf is located near where the sea stars were collected and recorded a mean temperature of 13.5 °C at 2 m depth in July 2020, the month with the warmest ocean temperatures. The target of 17 °C was 3.5 °C above that temperature. Temperature loggers (HOBO Pendant® MX2201, Onset Computer Corporation, Bourne, MA, USA) recorded water temperature once per minute in two temperature group tanks and one control group tank per repetition.

### 2.4. Statistical Analyses

Statistical analyses were performed using R (version 4.0.2) and the RStudio environment. For all analyses, a significance level of $p \leq 0.05$ was used. A Shapiro–Wilk test [45] was used to assess for normality; a parametric test was used for analytes with a Gaussian distribution, and a non-parametric test was used when samples had a non-Gaussian distribution.

A one-way Kruskal–Wallis test [46] was used to determine if there was a significant difference in each water quality parameter between the four groups during exposure. If a statistically significant difference was identified, a post hoc Dunn's test [47] was used to determine which groups differed. The *carb* function in the *seacarb* package [48] was used to evaluate carbonate chemistry in tank water from the pH and control groups. The measured pH and DIC were used as inputs to calculate $pCO_2$, $HCO_3$, $CO_3{}^2$, $\Omega_{Aragonite}$ and $\Omega_{Calcite.}$

A one-way Kruskal–Wallis test was used to determine if there were any differences in contoured diameter between the four groups at allocation to individual tanks, $T_0$, or $T_1$. Similarly, a one-way Kruskal–Wallis test was used to determine if there were any differences in wet weight between the four groups at allocation to individual tanks, baseline sampling, or sampling after exposure. If a statistically significant difference was identified, a post hoc Dunn's test was used to determine which groups differed.

To determine if there was a difference in coelomic fluid analytes between coelomic fluid samples collected at $T_0$ and $T_1$, a paired samples *t*-test test was performed for analytes with a Gaussian distribution, and a paired samples Wilcoxon signed-rank test was performed using the *wilcoxsign_test* function in the *coin* package for analytes with a non-Gaussian distribution. To determine if there was a difference in coelomic fluid analytes between each exposure group and the control group, for each exposure group and control group pair, a Welch two-sample *t*-test was performed for analytes with a Gaussian distribution, and a Wilcoxon rank sum test was performed using the *wilcox.test* function in the *stats* package for analytes with a non-Gaussian distribution. To determine if there was a difference in Mg, Na, K, Cl, Ca, and osmolality between randomly selected tank water samples and coelomic fluid samples at $T_0$ as well as $T_1$, a Wilcoxon rank sum test was performed using the *wilcox.test* function in the *stats* package.

## 3. Results

### 3.1. Animals

Sixty sea stars ($n$ = 15 per group) were initially enrolled into this study. One sea star in the hypoxia group was excluded after $T_0$ due to spine loss and lack of feeding. No samples from this individual were used in the present study. Another sea star in the hypoxia group autotomized two arms on day 11 of exposure and an additional arm on day 13 of exposure. At $T_1$ sampling, coelomic fluid was unable to be obtained, and the sea star was not included in the present study. The sea star was euthanized in an immersion of 75 g/L magnesium chloride [49] due to multiple autotomy, organ prolapse, and a poor prognosis.

Due to budget limitations, coelomic fluid analyses were conducted on a subset of individuals. Sea stars were randomly selected using a random number generator (Google, Mountain View, CA, USA). Coelomic fluid blood gas analyses were conducted on 39 sea stars each at $T_0$ and $T_1$ ($n$ = 10 from the pH, temperature, and control groups; $n$ = 9 from the hypoxia group). Coelomic fluid was analyzed for electrolytes, osmolality, and coelomocyte counts from 16 sea stars at $T_0$ ($n$ = 4 per group) and from 28 sea stars at $T_1$ ($n$ = 7 per group). All coelomic fluid samples from sea stars analyzed for electrolytes, osmolality, and coelomocyte counts also had blood gas analyses performed. Tank water electrolytes and osmolality were analyzed for seven tanks at $T_0$ ($n$ = 2 from the hypoxia, temperature and control groups; $n$ = 1 from the pH group) and for 13 tanks at $T_1$ ($n$ = 3 for the pH, temperature, and control groups; $n$ = 4 for the hypoxia group).

Sea stars had a mean diameter of 17.6 cm (range: 13.8–25.2 cm) and wet weight of 90.8 g (range: 50.6–166.9 g) when moved into individual tanks. At $T_0$, sea stars had a mean diameter of 17.8 cm (range: 15.0–26.0 cm) and wet weight of 97.4 g (range: 50.7–185.2 g). At $T_1$, sea stars had a mean diameter of 18.0 cm (range: 13.8–24.0 cm) and wet weight of 99.1 g (range: 43.3–196.3 g). There were no differences in contoured diameter or wet weight between the groups when moved into individual tanks or at $T_0$. There were statistically significant differences in both contoured diameter ($p$ = 0.032) and wet weight ($p$ = 0.020) at $T_1$. In post hoc testing, contoured diameter was significantly longer ($p$ = 0.012) in the hypoxia group (mean: 19.1 cm) compared to the temperature group (mean: 17.0 cm). Wet weight was significantly higher ($p$ = 0.002) in the hypoxia group (mean: 112.5 g) compared to the temperature group (mean: 82.6 g).

Figure 2 shows examples of sea stars from various treatment groups with normal or abnormal appearances during stressor exposure (between $T_0$ and $T_1$). No sea stars in the pH group were recorded as having abnormal behavior (Figure 2a). All sea stars in the hypoxia group were recorded as abnormal, with the sea stars having an inflated appearance, twisted arms, and decreased body turgor when handled for feeding (Figure 2d). Sea stars in the hypoxia group began exhibiting abnormal behaviors on average on day 7 of exposure (range: day 5–8). In the temperature group, two sea stars were recorded as abnormal during the ramp-up period: one sea star had an abnormal contour of the central disc on day 5 (12 °C), and the other had a deflated appearance on day 8 (15 °C). Both sea stars appeared grossly normal the following day. On day 4 and 5 of increased temperature exposure (17 °C), another sea star was noted to have an abnormal body wall contour over the central disc and arm. On day 5 through 9 of exposure, the sea star that had a deflated appearance during ramp-up was noted to have an inflated appearance (Figure 2e). One sea star in the temperature group was noted to intermittently have an abnormal body wall contour: during the ramp-up on day 6 (13 °C) through day 2 of exposure (17 °C), days 7–10 of exposure and on days 13–14 of exposure (Figure 2c). On day 9 of the ramp-up period (16 °C), a small amount of pyloric ceca was visible from the aboral surface of the distal tip of two arms (Figure 2b). The pyloric ceca was no longer visible 4 days later. No sea stars in the control group were recorded as abnormal.

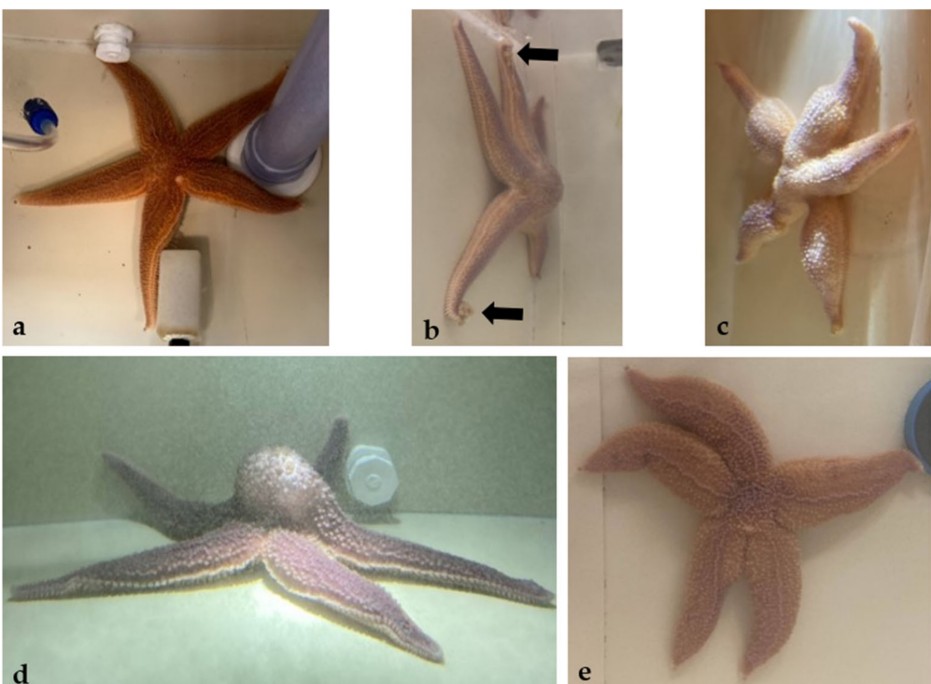

**Figure 2.** Photographs of adult common sea stars (*Asterias rubens*) taken during the time frame of 14 days of exposure to environmental stressors: (**a**) normal appearance and behavior (pH group); (**b**) sea star with pyloric ceca protruding from arm tips (solid arrows) (temperature group); (**c**) sea star with abnormal body wall contour on arms (temperature group); (**d**) sea star with abnormal body wall contour focused on central disc (hypoxia group); (**e**) sea star with inflated appearance (temperature group).

### 3.2. Water Quality

Ammonia was maintained between 0.00–0.130 mg/L, nitrite between 0.000–0.350 mg/L, nitrate between 0.8–6.3 mg/L, alkalinity between 6.4–7.7 dKH, and salinity between 31.72–34.18 ppt. The tank pH, DO, and temperature are reported in Table 1. Temperature logger results are reported in Table 2. Carbonate chemistry results for the pH and control groups are reported in Table 3. As expected, DIC and $pCO_2$ were higher in the tanks with decreased water pH.

**Table 1.** Water quality testing results for common sea stars (*Asterias rubens*, $n = 60$) individually housed for environmental exposures. A Kruskal–Wallis test was used to evaluate if there were differences between groups for each water quality parameter. A post hoc Dunn's test was performed to determine which groups differed for each statistically significant difference. * Denotes statistical significance.

| Parameter (Unit) | Treatment Group | A Priori Testing | | | | Post Hoc Testing | | |
| --- | --- | --- | --- | --- | --- | --- | --- | --- |
| | | Mean | Range | SD | *p* | Group Pairing | Difference in Means | *p* |
| pH | pH | 7.37 | 7.20–7.56 | 0.07 | <0.001 * | pH–Hypoxia | 0.56 | <0.001 * |
| | Hypoxia | 7.94 | 7.74–8.87 | 0.12 | | pH–Temperature | 0.48 | <0.001 * |
| | Temperature | 7.85 | 7.68–8.03 | 0.07 | | Temperature–Hypoxia | 0.09 | <0.001 * |
| | Control | 7.80 | 7.68–7.95 | 0.06 | | Control–pH | 0.44 | <0.001 * |
| | | | | | | Control–Hypoxia | 0.13 | <0.001 * |
| | | | | | | Control–Temperature | 0.04 | <0.001 * |

**Table 1.** *Cont.*

| Parameter (Unit) | Treatment Group | A Priori Testing | | | | Post Hoc Testing | | |
|---|---|---|---|---|---|---|---|---|
| | | Mean | Range | SD | *p* | Group Pairing | Difference in Means | *p* |
| Dissolved Oxygen (mg $O_2$/L) | pH | 9.41 | 9.05–9.73 | 0.12 | <0.001 * | pH–Hypoxia | 7.62 | <0.001 * |
| | Hypoxia | 1.80 | 1.12–5.25 | 0.34 | | pH–Temperature | 1.60 | <0.001 * |
| | Temperature | 7.81 | 7.56–8.04 | 0.09 | | Temperature–Hypoxia | 6.01 | <0.001 * |
| | Control | 9.49 | 8.13–9.74 | 0.15 | | Control–pH | 0.09 | <0.001 * |
| | | | | | | Control–Hypoxia | 7.70 | <0.001 * |
| | | | | | | Control–Temperature | 1.69 | <0.001 * |
| Temperature (°C) | pH | 7.7 | 7.3–8.0 | 0.2 | <0.001 * | pH–Hypoxia | 0.2 | <0.001 * |
| | Hypoxia | 7.5 | 6.9–7.7 | 0.2 | | pH–Temperature | 9.5 | <0.001 * |
| | Temperature | 17.2 | 16.5–17.5 | 0.2 | | Temperature–Hypoxia | 9.7 | <0.001 * |
| | Control | 7.6 | 6.8–8.0 | 0.2 | | Control–pH | 0.1 | <0.001 * |
| | | | | | | Control–Hypoxia | 0.2 | <0.001 * |
| | | | | | | Control–Temperature | 9.5 | <0.001 * |

**Table 2.** Tank temperature readings from temperature loggers recording once per minute for adult common sea stars (*Asterias rubens*) during exposures to increased temperature ("Temp", *n* = 6) and those in control group ("Control", *n* = 3). The target temperature for the temperature group was 18 °C, a 10 °C increase over the control group maintained at 8 °C.

| Tank | Mean Temperature (°C) | Temperature Range (°C) | SD |
|---|---|---|---|
| Temp 1 | 17.55 | 17.20–18.74 | 0.08 |
| Temp 2 | 17.82 | 17.54–18.66 | 0.10 |
| Temp 3 | 17.77 | 17.41–19.04 | 0.15 |
| Temp 4 | 17.76 | 13.74–18.10 | 0.23 |
| Temp 5 | 17.73 | 17.50–18.79 | 0.06 |
| Temp 6 | 17.88 | 17.67–18.70 | 0.05 |
| Control 1 | 8.06 | 7.68–8.960 | 0.14 |
| Control 2 | 8.13 | 7.81–8.88 | 0.15 |
| Control 3 | 7.95 | 6.78–8.960 | 0.28 |

**Table 3.** The mean ± standard deviation of water carbonate chemistry parameters measured (pH, alkalinity, dissolved inorganic carbon (DIC)) and calculated (all others) for pH group and control group tanks before ramp-up, after ramp-up, after 7 days of exposure and after 14 days of exposure.

| Timepoint | Group | pH | Alkalinity (mmol/kg) | DIC (mmol/kg) | $pCO_2$ (µatm) | $HCO_3^-$ (mmol/kg) | $CO_3^{2-}$ (mmol/kg) | $\Omega_{Aragonite}$ | $\Omega_{Calcite}$ |
|---|---|---|---|---|---|---|---|---|---|
| Before Ramp-Up | pH | 7.78 ± 0.09 | 1.64 ± 0.44 | 2.18 ± 0.10 | 783.5 ± 177.7 | 2.07 ± 0.10 | 0.07 ± 0.01 | 1.03 ± 0.17 | 1.63 ± 0.26 |
| | Control | 7.85 ± 0.04 | 1.66 ± 0.38 | 2.18 ± 0.12 | 655.6 ± 88.0 | 2.07 ± 0.11 | 0.08 ± 0.01 | 1.19 ± 0.11 | 1.89 ± 0.17 |
| After Ramp-Up | pH | 7.45 ± 0.05 | 1.50 ± 0.49 | 2.22 ± 0.12 | 1653.9 ± 181.3 | 2.11 ± 0.12 | 0.03 ± 0.01 | 0.49 ± 0.07 | 0.78 ± 0.11 |
| | Control | 7.82 ± 0.10 | 1.34 ± 0.47 | 2.14 ± 0.10 | 704.6 ± 172.9 | 2.03 ± 0.10 | 0.07 ± 0.01 | 1.10 ± 0.20 | 1.74 ± 0.32 |
| 7 days of Exposure | pH | 7.33 ± 0.06 | 1.35 ± 0.38 | 2.17 ± 0.12 | 2158.6 ± 206.9 | 2.05 ± 0.12 | 0.02 ± 0.004 | 0.36 ± 0.07 | 0.56 ± 0.11 |
| | Control | 7.79 ± 0.03 | 1.66 ± 0.63 | 2.05 ± 0.15 | 714.0 ± 87.9 | 1.95 ± 0.14 | 0.06 ± 0.01 | 0.97 ± 0.07 | 1.54 ± 0.11 |
| 14 days of Exposure | pH | 7.45 ± 0.05 | 1.39 ± 0.48 | 2.06 ± 0.17 | 1548.2 ± 273.2 | 1.96 ± 0.16 | 0.03 ± 0.003 | 0.45 ± 0.04 | 0.72 ± 0.06 |
| | Control | 7.82 ± 0.06 | 1.21 ± 0.40 | 2.00 ± 0.14 | 655.0 ± 107.0 | 1.90 ± 0.13 | 0.07 ± 0.01 | 1.02 ± 0.15 | 1.61 ± 0.23 |

### 3.3. Blood Gas Comparison by Exposure Group

Descriptive statistics for the blood gas comparison between $T_0$ and $T_1$ by group are reported in Table 4. In the pH group, sea star coelomic fluid had significantly lower $pH_{(TC)}$ (*p* = 0.011) and higher $pO_{2(TC)}$ (*p* = 0.013) when compared to $T_0$. Sea stars in the hypoxia and temperature groups had significant changes to all blood gas analytes when compared to $T_0$. In the hypoxia group, coelomic fluid had higher $pH_{(TC)}$ (*p* < 0.001) and $HCO_3$ (*p* < 0.001) as well as a significantly lower $pCO_{2(TC)}$ (*p* < 0.001) and $pO_{2(TC)}$ (*p* = 0.008) when compared to $T_0$. Coelomic fluid in the temperature group had significantly lower $pH_{(TC)}$ (*p* < 0.001), $pO_{2(TC)}$ (*p* < 0.001), and $HCO_3$ (*p* < 0.001) as well as a significantly higher

$pCO_{2(TC)}$ ($p < 0.001$) when compared to $T_0$. Sea stars in the control group had significantly higher $pCO_{2(TC)}$ ($p = 0.019$) when compared to $T_0$.

### 3.4. Electrolytes and Osmolality Comparison by Exposure Group

Descriptive statistics for the electrolytes and osmolality comparison between $T_0$ and $T_1$ by group are reported in Table 4. At $T_1$, sea stars in the pH group had significantly lower Ca ($p = 0.033$) and osmolality ($p = 0.004$); there were no significant changes in the hypoxia group. The temperature group had significantly lower Na ($p = 0.009$), K ($p = 0.023$), Cl ($p = 0.015$), and Ca ($p = 0.037$), and the control group had significantly lower K ($p = 0.004$) when compared to $T_0$.

### 3.5. Coelomocyte Count Comparison by Exposure Group

Descriptive statistics for coelomocyte count comparison by group are reported in Table 4. Coelomocyte clumps were observed in all samples; clumps were not counted. In all samples, 100% of the observed coelomocytes were the mononuclear phagocyte type [50]. There were no significant differences between coelomocyte counts between $T_0$ and $T_1$ for any groups. However, coelomocyte counts were trending downward in the pH group and upward in the hypoxia and temperature groups.

### 3.6. Comparison of Coelomic Fluid Analytes between Each Exposure Group and Control Group

Descriptive statistics for the comparison of coelomic fluid analytes between exposure groups and the control group at $T_1$ are reported in Table 5. Sea stars in the pH group had lower coelomic fluid $pH_{(TC)}$ ($p < 0.001$), $HCO_3$ ($p = 0.001$), and coelomocyte counts ($p = 0.011$) than sea stars in the control group. Sea stars in the hypoxia group had higher coelomic fluid $pH_{(TC)}$ ($p = 0.004$), lower $pCO_{2(TC)}$ ($p < 0.001$) and $pO_{2(TC)}$ ($p < 0.001$), and higher coelomocyte counts ($p = 0.002$) compared to the control group. Sea stars in the temperature group had lower coelomic fluid $pH_{(TC)}$ ($p = 0.002$), higher $pCO_{2(TC)}$ ($p < 0.001$), lower $pO_{2(TC)}$ ($p < 0.001$), and higher coelomocyte counts ($p = 0.002$) compared to the control group.

### 3.7. Coelomic Fluid Electrolytes and Osmolality Compared to Tank Water

Descriptive statistics for the comparison of coelomic fluid and tank water electrolytes and osmolality are reported in Table 6. Sea stars from all groups were combined for $T_0$ due to small sample size. At $T_0$, sea star coelomic fluid had higher K ($p = 0.017$) and osmolality ($p = 0.018$) compared to tank water. At $T_1$ analysis, sea stars in the hypoxia group had higher coelomic fluid K ($p = 0.028$) compared to tank water. There were no statistically significant differences between coelomic fluid in hypoxia, pH, and control groups and tank water electrolytes and osmolality.

**Table 4.** Mean ± standard deviation for blood gases (*n* = 10 animals each in pH, temperature; and control groups; hypoxia group had 9 animals), electrolytes (*n* = 4 animals per group), osmolality (*n* = 4 animals per group) and coelomocyte counts (*n* = 4 animals per group) in coelomic fluid of common sea stars (*Asterias rubens*) exposed to 14 days ($T_1$) of decreased pH (−0.4 units), hypoxia (target dissolved oxygen ~1.75 mg $O_2$/L), increased temperature (+10 °C) and controls compared to baseline measurements ($T_0$) from the same individual. Abbreviations: BLD, below limit of detection; D, distribution; G, Gaussian; NG, non-Gaussian; TC, temperature correction [40]; $TCO_2$, total carbon dioxide. * Denotes statistical significance. Background indicates statistically significant results.

| | Group | | | | | | | | | | | | | | | | |
|---|---|---|---|---|---|---|---|---|---|---|---|---|---|---|---|---|---|
| Analyte | pH | | | | Hypoxia | | | | Temperature | | | | Control | | | |
| | $T_0$ Mean | $T_1$ Mean | *p* | D | $T_0$ Mean | $T_1$ Mean | *p* | D | $T_0$ Mean | $T_1$ Mean | *p* | D | $T_0$ Mean | $T_1$ Mean | *p* | D |
| pH$_{(TC)}$ | 7.42 ± 0.08 | 7.31 ± 0.08 | 0.011 * | G | 7.45 ± 0.10 | 7.65 ± 0.07 | <0.001 * | G | 7.56 ± 0.06 | 7.35 ± 0.08 | <0.001 * | G | 7.54 ± 0.05 | 7.51 ± 0.11 | 0.426 | G |
| pCO$_{2(TC)}$ (mmHg) | 2.2 ± 0.2 | 2.4 ± 0.2 | 0.059 | NG | 2.1 ± 0.2 | 1.8 ± 0.2 | <0.001 * | G | 2.1 ± 0.2 | 3.0 ± 0.2 | <0.001 * | G | 2.0 ± 0.1 | 2.3 ± 0.2 | 0.019 * | G |
| pO$_{2(TC)}$ (mmHg) | 103 ± 9 | 113 ± 6 | 0.013 * | G | 107 ± 6 | 55 ± 2 | 0.008 * | NG | 114 ± 8 | 98 ± 7 | <0.001 * | G | 113 ± 6 | 112 ± 8 | 0.455 | G |
| HCO$_3$ (mmol/L) | 1.9 ± 0.4 | 1.6 ± 0.3 | 0.103 | NG | 1.9 ± 0.3 | 2.7 ± 0.4 | <0.001 * | G | 2.4 ± 0.2 | 2.0 ± 0.3 | <0.001 * | G | 2.2 ± 0.3 | 2.4 ± 0.5 | 0.231 | G |
| TCO$_2$ (mmol/L) | BLD | BLD | - | - | BLD | BLD | - | - | BLD | BLD | - | - | BLD | BLD | - | - |
| Lactate (mmol/L) | BLD | BLD | - | - | BLD | BLD | - | - | BLD | BLD | - | - | BLD | BLD | - | - |
| Magnesium (mmol/L) | 52 ± 3 | 54 ± 7 | 0.690 | G | 51 ± 6 | 47 ± 6 | 0.201 | G | 65 ± 13 | 54 ± 4 | 0.068 | NG | 51 ± 7 | 54 ± 2 | 0.564 | G |
| Sodium (mmol/L) | 427 ± 6 | 416 ± 10 | 0.148 | G | 425 ± 3 | 402 ± 57 | 0.715 | NG | 451 ± 8 | 435 ± 4 | 0.009 * | G | 442 ± 4 | 432 ± 4 | 0.111 | G |
| Potassium (mmol/L) | 9.4 ± 0.5 | 9.2 ± 0.3 | 0.294 | G | 9.4 ± 0.1 | 9.5 ± 1.0 | 0.886 | G | 10.0 ± 0.1 | 9.6 ± 0.3 | 0.023 * | G | 9.8 ± 0.1 | 9.4 ± 0.1 | 0.004 * | G |
| Chloride (mmol/L) | 495 ± 7 | 482 ± 12 | 0.118 | G | 492 ± 4 | 468 ± 64 | 0.715 | NG | 521 ± 8 | 505 ± 4 | 0.015 * | G | 511 ± 5 | 501 ± 7 | 0.144 | G |
| Calcium (mmol/L) | 9.7 ± 0.1 | 9.4 ± 0.2 | 0.033 * | G | 9.5 ± 0.2 | 9.2 ± 1.1 | 0.853 | NG | 10.0 ± 0.1 | 9.8 ± 0.1 | 0.037 * | G | 9.9 ± 0.3 | 9.6 ± 0.2 | 0.199 | G |
| Osmolality (mmol/kg) | 959 ± 8 | 941 ± 4 | 0.004 * | G | 944 ± 4 | 939 ± 6 | 0.138 | G | 984 ± 5 | 968 ± 25 | 0.232 | G | 986 ± 6.6 | 944 ± 3.4 | 0.066 | G |
| Coelomocyte Counts (× 10$^3$ cells/Ml) | 15.59 ± 15.88 | 5.78 ± 7.07 | 0.068 | NG | 8.90 ± 13.60 | 14.10 ± 22.15 | 0.144 | NG | 8.58 ± 12.43 | 11.78 ± 14.66 | 0.068 | NG | 1.14 ± 0.77 | 1.13 ± 0.37 | 0.971 | NG |

**Table 5.** Comparison of common sea star (*Asterias rubens*) coelomic fluid blood gases (*n* = 10 animals each in pH, temperature and control groups, hypoxia group had 9 animals), electrolytes (*n* = 7 animals per group), osmolality (*n* = 7 animals per group) and coelomocyte counts (*n* = 7 animals per group) between each treatment group versus the control group after 2 weeks of exposure ($T_1$). For analytes where both groups had a Gaussian distribution (G), a Welch's two sample t-test was used. For analytes where one or both groups had a non-Gaussian distribution (NG), a Wilcoxon rank sum test was performed. Abbreviations: BLD, below limit of detection; D, distribution; TC, temperature correction [40]; $TCO_2$, total carbon dioxide. * Denotes statistical significance. Background indicates statistically significant results.

| Analyte | Control Group $T_1$ Mean ± SD | D | pH Group $T_1$ Mean ± SD | p | D | Hypoxia Group $T_1$ Mean ± SD | p | D | Temperature Group $T_1$ Mean ± SD | p | D |
|---|---|---|---|---|---|---|---|---|---|---|---|
| | | | **pH vs. Control** | | | **Hypoxia vs. Control** | | | **Temperature vs. Control** | | |
| pH(TC) | 7.51 ± 0.11 | G | 7.31 ± 0.08 | <0.001 * | G | 7.65 ± 0.07 | 0.004 * | G | 7.35 ± 0.07 | 0.002 * | G |
| pCO2(TC) (mmHg) | 2.3 ± 0.2 | G | 2.4 ± 0.2 | 0.540 | NG | 1.8 ± 0.2 | <0.001 * | G | 3.0 ± 0.2 | <0.001 * | G |
| pO2(TC) (mmHg) | 112 ± 8 | G | 113 ± 6 | 0.638 | G | 55 ± 2 | <0.001 * | NG | 98 ± 7 | <0.001 * | G |
| HCO3 (mmol/L) | 2.4 ± 0.5 | G | 1.6 ± 0.3 | 0.001 * | NG | 2.7 ± 0.4 | 0.182 | G | 2.0 ± 0.3 | 0.066 | G |
| TCO2 (mmol/L) | BLD | - | BLD | BLD | - | BLD | BLD | - | BLD | BLD | - |
| Lactate (mmol/L) | BLD | - | BLD | BLD | - | BLD | BLD | - | BLD | BLD | - |
| Magnesium (mmol/L) | 56 ± 4 | G | 54 ± 1 | 0.311 | G | 52 ± 9 | 0.293 | G | 54 ± 5 | 0.339 | G |
| Sodium (mmol/L) | 438 ± 10 | G | 426 ± 16 | 0.109 | G | 419 ± 46 | 0.443 | NG | 438 ± 10 | 0.919 | G |
| Potassium (mmol/L) | 9.6 ± 0.3 | G | 9.4 ± 0.3 | 0.230 | G | 9.5 ± 0.7 | 0.756 | G | 9.9 ± 0.6 | 0.193 | G |
| Chloride (mmol/L) | 508 ± 11 | G | 493 ± 18 | 0.087 | G | 486 ± 51 | 0.609 | NG | 508 ± 10 | 0.982 | G |
| Calcium (mmol/L) | 9.8 ± 0.3 | G | 9.5 ± 0.3 | 0.072 | G | 9.5 ± 0.9 | 0.949 | NG | 9.8 ± 0.2 | 0.999 | G |
| Osmolality (mmol/kg) | 962 ± 23 | G | 955 ± 19 | 0.566 | G | 956 ± 22 | 0.686 | G | 983 ± 25 | 0.130 | G |
| Coelomocyte Counts (×$10^3$ cells/µL) | 1.04 ± 0.29 | G | 3.96 ± 5.49 | 0.011 * | NG | 9.41 ± 16.72 | 0.002 * | NG | 7.99 ± 11.41 | 0.002 * | NG |

**Table 6.** Common sea star (*Asterias rubens*) coelomic fluid [SeaStar] compared to tank water [Tank] electrolyte and osmolality using a Wilcoxon rank sum test. Coelomic fluid and tank water samples were paired for analysis; the sample size (*n*) given represents the number of individuals for each type of analysis. Statistical significance was determined with $p < 0.05$. Sea stars were randomly assigned to one of four groups: decreased pH ($-0.4$ units), hypoxia (target dissolved oxygen ~1.75 mg $O_2$/L), increased temperature ($+10\ ^\circ$C) or control. Samples were taken at baseline ($T_0$) and after exposure for 14 days ($T_1$). * Denotes statistical significance. Background indicates statistically significant results.

| | | | | Group | | | | | | | |
| | | | | pH | | Hypoxia | | Temperature | | Control | |
| Analyte (Unit) | Sample Type | $T_0$ Mean (*n* = 14) | *p* | $T_1$ Mean (*n* = 3) | *p* | $T_1$ Mean (*n* = 4) | *p* | $T_1$ Mean (*n* = 3) | *p* | $T_1$ Mean (*n* = 3) | *p* |
|---|---|---|---|---|---|---|---|---|---|---|---|
| Magnesium (mmol/L) | SeaStar | 54 | 0.311 | 51 | 0.200 | 49 | 0.715 | 51 | 0.166 | 57 | 0.100 |
| | Tank | 46 | | 42 | | 47 | | 47 | | 50 | |
| Sodium (mmol/L) | SeaStar | 338 | 0.535 | 428 | 0.100 | 435 | 0.200 | 441 | 0.700 | 440 | 0.100 |
| | Tank | 394 | | 337 | | 409 | | 410 | | 401 | |
| Potassium (mmol/L) | SeaStar | 9.6 | 0.017 * | 9.4 | 0.100 | 9.6 | 0.028 * | 10.0 | 0.109 | 9.8 | 0.100 |
| | Tank | 8.2 | | 7.0 | | 8.6 | | 8.6 | | 8.4 | |
| Chloride (mmol/L) | SeaStar | 506 | 0.445 | 496 | 0.100 | 504 | 0.057 | 512 | 0.593 | 509 | 0.100 |
| | Tank | 452 | | 391 | | 452 | | 476 | | 471 | |
| Calcium (mmol/L) | SeaStar | 9.8 | 0.203 | 9.6 | 0.200 | 9.9 | 0.068 | 9.9 | 0.100 | 9.9 | 0.166 |
| | Tank | 8.9 | | 7.8 | | 8.8 | | 9.2 | | 9.1 | |
| Osmolality (mmol/kg) | SeaStar | 969 | 0.018 * | 963 | 0.200 | 962 | 0.068 | 975 | 0.400 | 969 | 1.00 |
| | Tank | 866 | | 824 | | 911 | | 944 | | 967 | |

## 4. Discussion

This study provides new information on the changes in sea star coelomic fluid in response to short-term exposure to environmental stressors consistent with future oceanic conditions anticipated with climate change.

Point-of-care blood gas analysis has not been previously reported in sea stars. This technique provided novel and useful information in evaluating the respiratory gases and acid–base balance of sea star coelomic fluid. Care must be taken in the interpretation due to limited data about blood gas evaluation in invertebrates and the importance of considering them as poikilotherms for which correction of blood gas results for tank water temperature is required. Base excess and oxygen saturation ($sO_2$) were excluded from this study since their quantifications are based on human hemoglobin and plasma proteins as well as oxygen affinity of human blood, respectively [41]. Baseline coelomic fluid $pH_{(TC)}$ reported by the iSTAT analyzer ranged from 7.30–7.65, which is to be expected as echinoderm coelomic fluid pH is 0.5–1.5 pH units lower than seawater. This pH difference may be due to $CO_2$ diffusion rates and acidic metabolites [51].

Blood gas analysis results showed substantial changes in respiratory gas exchange and acid–base balance in sea stars after stressor exposure. The decreased coelomic fluid $pH_{(TC)}$ in the pH group and temperature group between $T_0$ and $T_1$ was likely driven by the increase in coelomic fluid $pCO_{2(TC)}$. In the pH group, this $pCO_{2(TC)}$ increase may have resulted from the increased $pCO_2$ in the water, as bubbled $CO_2$ was used to decrease tank water pH. This is consistent with what was previously reported in common sea stars experimentally exposed to decreased pH [52]. In the temperature group, the increase in coelomic fluid $pCO_{2(TC)}$ suggests increased respiration due to increased metabolism. Intertidal dwarf cushion sea stars (*Parvulastra exigua*) had increased metabolism due to increased water temperature, while decreased pH had no impact on metabolism [53]. A similar increase in metabolic rates with increasing water temperatures was reported in Pacific crown-of-thorns sea stars (*Acanthaster* sp.) [27]. In both the pH and temperature groups, sea stars appeared to attempt to compensate the decreased coelomic fluid pH, as evidenced by decreased bicarbonate. The buffering capacity of echinoderm coelomic fluid is predominantly (94%) due to the bicarbonate buffer system of seawater as well as a small contribution by coelomocytes through an unknown mechanism [54].

The increase in coelomic fluid $pH_{(TC)}$ in the hypoxia group when compared to $T_0$ was driven by increased bicarbonate combined with a decrease in $pCO_{2(TC)}$. The bubbling of nitrogen to decrease DO increased the pH of the tank by approximately 0.13 units compared to that of the control group. The mean coelomic fluid $pH_{(TC)}$ increased by an average of 0.20 units, indicating this may be a response to the hypoxic conditions, rather than solely to the slight change in tank pH due to nitrogen gas. A similar increase in coelomic fluid pH and bicarbonate and decrease in $pCO_2$ were observed in green sea urchins (*Psammechinus miliaris*) exposed to hypoxic conditions, with no change in tank water pH [55]. Although the cause of the increased pH and bicarbonate is unknown, it may be due to by-products of anaerobic metabolism, as has been reported for peanut worms (*Sipunculus nudus*) [56]. However, the ability of echinoderms for anaerobic metabolism is not well studied [57]. The decreased $pCO_2$ was likely due to decreased respiration rate rather than metabolic suppression, as purple urchins (*Strongylocentrotus purpuratus*) exposed to hypoxia had stable baseline metabolic rates [58].

The oxygenation status of the sea stars was impacted by short-term environmental stressor exposure. Sea stars in the hypoxia and temperature groups had decreased oxygenation status between $T_0$ and $T_1$ with decreased coelomic fluid $pO_{2(TC)}$. The $pO_{2(TC)}$ was markedly decreased in the hypoxia group, from 107 to 55 mmHg, indicating that sea stars were unable to regulate coelomic fluid oxygen in the face of hypoxia. This is consistent with findings that green sea urchins (*P. miliaris*) were also unable to regulate coelomic fluid oxygen in experimentally induced hypoxia [55]. The decreased oxygenation status in the increased temperature group was likely influenced by both decreased tank water DO and increased metabolic demand. There is an inverse relationship between DO

and water temperature led by the decreased solubility of oxygen in water as temperature increases [59]. Oxygen consumption increased with increasing water temperature in dwarf cushion sea stars (*P. exigua*) [53]. Sea stars in the decreased pH group had increased coelomic fluid $pO_{2(TC)}$. This may be due to an increased efficiency of gas exchange under those physiological conditions or decreased consumption due to metabolic suppression; however, no change in metabolism due to exposure to decreased water pH was observed in dwarf cushion sea stars (*P. exigua*) [53].

Evaluation of coelomic fluid electrolytes and osmolality provided insight into effects of stressors on basic metabolic processes in this study. Few studies have examined the relationship of sea star coelomic fluid electrolytes to environmental stressors, other than studies investigating effects of hypo- and hyper-salinity [60,61]. Compared to $T_0$, sea stars in the pH group had hypo-osmolar coelomic fluid at $T_1$, driven by decreased Ca when compared to $T_0$. A previous study found no change in Mg and Ca concentrations in the coelomic fluid of common sea stars exposed to water pH of 7.4, 7.7 or 8.0 after 7 or 14 days. However, osmolality was not evaluated, and no baseline electrolyte concentrations were available for comparison [12]. This decrease in coelomic fluid Ca may be due to changes in the carbonate equilibrium of the tank water. The addition of $CO_2$ into seawater decreases the pH and changes the balance between bicarbonate and carbonate ions in the water [62]. As the carbonate equilibrium changes, carbonate ions are no longer bioavailable to marine organisms [63]. This is further supported by the finding of decreased tank water calcium concentrations in the pH group. The calcium concentration of the tank water in the pH group decreased from a mean of 8.9 mmol/L at $T_0$ to 7.8 mmol/L at $T_1$. Sea stars in the temperature group had decreased coelomic fluid Na, K, Cl, and Ca compared to $T_0$. There are no previously published studies evaluating the relationship between coelomic fluid electrolytes and increased water temperatures in echinoderms. These electrolyte changes may reflect that exposure to increased water temperature challenges osmoregulatory control mechanisms. The control group had lower coelomic fluid K compared to $T_0$ measurements, which may be due to differences in feeding times, behavior, or other undetermined factors impacting active K transport [64]. However, changes in coelomic fluid electrolytes should be interpreted with caution as results were not significantly different when compared to the control group or tank water, which suggests they may not be biologically relevant.

Coelomocyte clumping was observed in every sample and impacted the ability to detect differences in cell counts. Clumping of coelomocytes, similar to hemostasis in vertebrates, occurs rapidly to prevent loss of coelomic fluid following injuries to the body wall of sea stars [65]. In eight different sea star species, clumping occurred by coelomocyte aggregation [66]. In *Asterias forbesi*, coelomocyte clumping is dependent on calcium and magnesium and occurs in two phases [65]. Considerations for the degree of clumping observed in this study include delay in processing (i.e., 3 h before cytocentrifugation was possible), effects from temperature differences, or incomplete anticoagulation.

The small sample size may have obscured changes in coelomocyte counts between $T_0$ and $T_1$ sampling. Coelomocyte counts at $T_1$ did not differ in any group when compared to $T_0$ count; however, there were trends in coelomocyte counts in the groups that were not statistically significant, and comparison to the control group indicated coelomocyte counts were higher in the pH, hypoxia, and temperature groups. While the pH group had higher coelomocyte counts than the control group, coelomocyte counts were trending downward (37% decrease) from $T_0$ measurements, so the increase should be interpreted with caution. A 50% decrease in coelomocyte count was observed in *A. rubens* after 6 months of exposure to decreased pH (pH 7.7) [13]. As coelomocytes are the immune effector cell for echinoderms [33], a decrease in circulating coelomocytes indicates immunosuppression, which may result in common sea stars being more susceptible to infection with pathogens or opportunistic microorganisms. After 2 weeks of exposure to hypoxia or increased temperature, coelomocyte counts were trending upwards when compared to baseline counts. Coelomocyte counts did not change in sea stars after 3 days of exposure to hypoxia [67] or 1 week of exposure to increased temperature (+5 °C) [68], which may indicate that coelomo-

cyte responses lag behind environmental stressor exposure. Increased coelomocyte counts have been observed in ochre sea stars (*Pisaster ochraceus*) with clinical signs consistent with SSW [39,69] and indicate an inflammatory response, which comes at a systemic energetic cost to the sea star.

When each exposure group was compared to the control group at $T_1$, many of the identified coelomic fluid changes were consistent with the $T_0$ to $T_1$ comparison. One notable exception was that no changes in electrolytes were significant when compared to the control group. This may be due to the small sample size for the $T_0$ to $T_1$ comparison or the wide variability of coelomic fluid electrolytes between individuals. The coelomic fluid changes that were consistent between both the $T_0$ to $T_1$ and exposure to control group comparisons likely represent the most biologically relevant coelomic fluid responses. The pH group had lower $pH_{(TC)}$ in both comparisons ($T_0$ to $T_1$ and each exposure group compared to the control group), which indicates they were unable to adequately compensate for the decreased tank water pH and bring their coelomic fluid pH back to the pH at $T_0$. The hypoxia group had higher $pH_{(TC)}$, lower $pCO_{2(TC)}$ and $pO_{2(TC)}$, and higher coelomocyte counts in both comparisons ($T_0$ to $T_1$ and each exposure group compared to the control group). The higher $pH_{(TC)}$ may be due to products of anaerobic metabolism combined with decreased $pCO_{2(TC)}$ from decreased respiration. The lower $pO_{2(TC)}$ indicates that the sea stars could not regulate coelomic fluid $pO_2$ in hypoxic water conditions. When compared to the control group, the temperature group had a lower $pH_{(TC)}$ likely driven by a higher $pCO_{2(TC)}$ from increased metabolism, with a lower $HCO_3$ due to consumption of coelomic fluid buffering capacity. The temperature group also had a lower $pO_{2(TC)}$, likely due to the decreased solubility of oxygen in water as the temperature increased. In both the hypoxia and temperature groups, coelomocyte counts were increased compared to controls, which indicates inflammation.

Sea stars have historically been considered to be iso-ionic and iso-osmolar to the water that surrounds them. Some studies have reported higher K concentrations in the coelomic fluid of *A. rubens* compared to seawater, especially in the water vascular system [60,70]. A previous study reported that coelomic fluid of *A. rubens* was iso-osmotic to seawater [71]. However, a more recent study found that 14 species of sea stars from the Pacific Ocean were slightly hyper-osmotic to the seawater surrounding them [72]. This is consistent with the higher coelomic fluid K and osmolality observed at $T_0$. The increased K and osmolality suggest active ion regulation in coelomic fluid. At $T_1$, K was higher only in the hypoxia group. This may indicate the sea stars were struggling with K regulation; however, in all groups, coelomic fluid K remained higher than tank water K. The lack of statistical significance may be due to the small sample size.

Compensation for environmental stressors comes at an energetic cost, which may result in trade-offs for maintenance, reproduction, development, and growth [73]. This is exemplified by the differences in size, both in weight and contoured diameter, between the hypoxia and temperature groups at $T_1$. Sea stars in the hypoxia group also appeared grossly inflated, with spaces visible between ossicles and spines. In one study, the growth of common sea stars exposed to increased water temperatures (+4 °C) for 2 months was reduced significantly [29]. Thus, in the current study, the subjective inflation in the hypoxia group combined with decreased size in the increased temperature group may have resulted in the differences in morphometric measurements since there were no significant differences between the groups at allocation to individual tanks or baseline sampling. The finding of increased coelomic fluid $pCO_{2TC}$ in the temperature group further supports the likely impact of increased water temperature on metabolism and potentially growth.

This study has several limitations, including small sample size in the control and treatment groups, the ex situ nature of exposures, short duration of exposures, and high variability in some analytes. This study was intended to be conducted on a short time scale compared to projected global climate change due to logistical challenges. While every effort was made to use near-future ocean conditions, these actual changes are uncertain, and are projected to occur over several generations of sea stars, and adaptations may occur over

time. These projected oceanic changes will also not occur as isolated stressors, but rather in combination and may be synergistic or antagonistic in their combined impacts. However, it was important to evaluate each environmental stressor by itself to establish what the effects were before delving into the complicated interactions of multiple simultaneous stressors. The high variability of some analytes, as evidenced by the high standard deviation (e.g., magnesium and sodium), may have resulted in statistical differences due to chance rather than being representative as a response to environmental stressors. However, all statistically significant $T_1$ mean values fell outside of the range for $T_0$ mean $\pm$ standard deviation, thus supporting the observed statistical differences. There is a risk of false positives with multiple comparisons. A Bonferroni correction reduces the risk of Type I error at the expense of increasing Type II error rates and is no longer recommended [74]. Since none of the criteria for recommending a Bonferroni correction were met and to avoid an increase in Type II error rates, a correction for multiple comparisons was not conducted.

## 5. Conclusions

This study shows that sea stars have coelomic fluid responses to simulated environmental stressors indicative of systemic effects, including shifts in energy balance, effects on the cellular immune system, and respiratory or metabolic derangements that may result in decreased overall condition. This vulnerability to changes in oceanic conditions may provide some explanation for the increasing frequency and severity of sea star wasting events observed around the globe. Point-of-care blood gas analysis, electrolytes, osmolality, and coelomocyte counts can be helpful tools in evaluating the health status of both free-ranging and captive sea stars. These diagnostic tests routinely used in vertebrate animals can be accessible, minimally invasive methods to evaluate the response of invertebrates to environmental stressors. Ultimately, this study provides useful information about the coelomic fluid responses of sea stars to short-term environmental stressors that can be built upon to determine how sea stars may fare with changing oceanic conditions such as those projected to occur with climate change.

**Supplementary Materials:** The following supporting information can be downloaded at: https://www.mdpi.com/article/10.3390/fishes8010051/s1, Table S1: Chemistry assay methods and analytical precision applicable to sea star coelomic fluid and tank water based on quality control data.

**Author Contributions:** Conceptualization, S.J.W. and N.I.S.; Data curation, S.J.W.; Formal analysis, S.J.W.; Funding acquisition, S.J.W. and N.I.S.; Investigation, S.J.W., I.H. and N.I.S.; Methodology, S.J.W., K.J.K., D.C.B., J.E.A., B.W., A.L.N., K.E., I.H. and N.I.S.; Project administration, N.I.S.; Resources, K.J.K., D.C.B. and N.I.S.; Software, S.J.W.; Supervision, S.J.W. and N.I.S.; Validation, N.I.S.; Visualization, N.I.S.; Writing—original draft, S.J.W. and N.I.S.; Writing—review and editing, S.J.W., D.C.B., J.E.A., B.W., A.L.N., K.E. and I.H.. All authors have read and agreed to the published version of the manuscript.

**Funding:** This research was funded by a grant from the Wild Animal Health Fund of the American Association of Zoo Veterinarians (AAZV; Grant number 2020 #40) and financial support from ZooQuatic Laboratory, LLC.

**Institutional Review Board Statement:** Not applicable for non-cephalopod invertebrates. This research involved the use of non-lethal experimental procedures.

**Data Availability Statement:** The data presented in this study are available on request from the corresponding author.

**Acknowledgments:** The authors wish to thank Laurie Adler, Andrew Kane, Ross Brooks, Celine Dalrymple, Haley Diefenbaugh, Jamie Garde, Miranda Gibson, Makenzie Griffin, and Miranda Jackson for their assistance with sea star housing and exposures; and Nancy Denslow and Maria Byrne for sharing their knowledge.

**Conflicts of Interest:** B.W., J.E.A. and H.L.N are employed by ZooQuatic Laboratory LLC. The other authors declare no conflict of interest.

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
