# Peer review of "Common Sea Star (Asterias rubens) Coelomic Fluid Changes in Response to Short-Term Exposure to Environmental Stressors"

_fishes, doi:10.3390/fishes8010051_

Round 1
Reviewer 1 Report
The manuscript revealed the coelomic fluid changes of common sea star under short-term environmental stressors, including decreased pH, hypoxia and increased temperature. The results is clear presented and very interesting. The introduction in the second paragraph is much more about the sea star wasting syndrome. Please simplify it. Some details need rewrite, as show below:
1. Line 22 of article,“O2/L”:The number 2 should be changed to the subscript (“O2/L”).
2. The fonts of the numbers in the abstract are inconsistent.
3. Are “compare” on line 23 and “assessed”on line 19 parallel predicates? If they are, "to compare" should be changed to "compared".
4. In the full text, there are missing spaces between the numbers and the temperature units (℃).
5. Do some tests, such as A Shapiro-Wilk test (line 275), Aone-way Kruskal Wallis test (line 279), and a post-hoc Tukey's HSD test (line 281), need to add references in the statistical analysis? And the carb function in line 282 also need some reference.
6. The line numbers of lines 396-400 in the article overlap with Table 2, and line numbers of line 486 overlap with Table 6.
7. 412 line Table 3 title needs to be aligned (DIC (m mol/kg)).
8. The ”Fig” that appears in the text of the article is different from the actual name of the picture (“Figure”).
9. Does the reference format need to be unified? Now some “year” in bold, some not.
Author Response
The manuscript revealed the coelomic fluid changes of common sea star under short-term environmental stressors, including decreased pH, hypoxia and increased temperature. The results is clear presented and very interesting. The introduction in the second paragraph is much more about the sea star wasting syndrome. Please simplify it.
RESPONSE: THANK YOU FOR YOUR POSITIVE FEEDBACK. WE HAVE DELETED THE SENTENCES ON LINES 42-44 TO SIMPLIFY THE PARAGRAPH ABOUT SEA STAR WASTING SYNDROME. THAT PARAGRAPH WAS INCLUDED TO HIGHLIGHT THE NEED FOR THIS RESEARCH.
Some details need rewrite, as show below:
- Line 22 of article,“O2/L”:The number 2 should be changed to the subscript (“O2/L”).
REPONSE: CHANGE MADE. THIS CHANGE WAS ALSO MADE ON LINE 115.
- The fonts of the numbers in the abstract are inconsistent.
RESPONSE: THE NUMBERS IN LINE 23 HAVE BEEN CHANGED TO PALATINO LINOTYPE.
- Are “compare” on line 23 and “assessed” on line 19 parallel predicates? If they are, "to compare" should be changed to "compared".
RESPONSE: THEY ARE PARALLEL PREDICATES AND “TO COMPARE” HAS BEEN CHANGED TO “COMPARED” IN LINES 23 AND 116.
- In the full text, there are missing spaces between the numbers and the temperature units (℃).
RESPONSE: SPACES HAVE BEEN ADDED BETWEEN NUMBERS AND THE TEMPERATURE UNITS THROUGHOUT THE TEXT.
- Do some tests, such as A Shapiro-Wilk test (line 275), A one-way Kruskal Wallis test (line 279), and a post-hoc Tukey's HSD test (line 281), need to add references in the statistical analysis? And the carb function in line 282 also need some reference.
RESPONSE: CITATIONS WERE ADDED TO LINE 282, 286, 288, AND 289. THE FOLLOWING REFERENCES WERE ADDED (IN ORDER OF APPEARANCE):
Shapiro, S.S.; Wilk, M.B. An Analysis of Variance Test for Normality (Complete Samples). Biometrika 1965, 52, 591, doi:10.2307/2333709.
Kruskal, W.H.; Wallis, W.A. Use of Ranks in One-Criterion Variance Analysis. J. Am. Stat. Assoc. 1952, 47, 583–621, doi:10.1080/01621459.1952.10483441.
Tukey, J.W. Comparing Individual Means in the Analysis of Variance. Biometrics 1949, 5, 99–114.
Gattuso, J.P.; Lavigne, H. Technical Note: Approaches and software tools to investigate the impact of ocean acidification. Biogeosciences 2009, 6, 2121–2133, doi:10.5194/bg-6-2121-2009.
- The line numbers of lines 396-400 in the article overlap with Table 2, and line numbers of line 486 overlap with Table 6.
RESPONSE: THE OVERLAPS WITH LINE NUMBER 396-400 AND TABLE 2 AS WELL AS LINE 489 AND TABLE 6 HAVE BEEN CORRECTED.
- 412 line Table 3 title needs to be aligned (DIC (m mol/kg)).
RESPONSE: CHANGE MADE.
- The ”Fig” that appears in the text of the article is different from the actual name of the picture (“Figure”).
RESPONSE: “Fig” HAS BEEN CHANGED TO “Figure” AT LINES 344, 346, 354, 357, AND 359.
- Does the reference format need to be unified? Now some “year” in bold, some not.
RESPONSE: IN THE INSTRUCTIONS FOR AUTHORS (AVAILABLE AT https://www.mdpi.com/journal/fishes/instructions#references) YEAR IS BOLDED FOR JOURNAL ARTICLES AND NOT FOR BOOKS OR BOOK CHAPTERS; THIS IS THE DIFFERENCE IN BOLDING SEEN IN OUR REFERENCE LIST. REFERENCES WERE PREPARED USING MENDELEY AS RECOMMENDED.
Reviewer 2 Report
Comments to the manuscript ’Common sea star (Asterias rubens) coelomic fluid changes in response to short-term exposure to environmental stressors’ by Wahltinez et al.
The authors investigated the effect of changes in pH, oxygen concentration and temperature on coelomic parameters in the sea stars. The manuscript may end up being publishable after serious revision, but I have questions to the study design, some of the experimental procedures, the description of the experimental procedures, the presentation and interpretation of data, the statistical treatment and to the data themselves.
Study design and experimental procedure:
The authors take out samples from their experimental animals before ramp up of the parameters investigated and they ramp down after the T1 coelomic fluid sampling. The authors do not explain why they carry out this ramp down after the apparent finalisation of the experiment. Since they carried out this ramp down, it would have been very useful if they had taken out coelomic fluid samples after the ramp down to see if coelomic fluid parameters had returned to T0-values.
The authors choose to analyse coelomic parameters for a subset of the experimental animals, a procedure which I find very questionable. The authors argue that it has been done to minimise experimental costs but, in my experience, the factual costs of performing the analyses carried out are not very heavy for the individual analysis. The authors selected different numbers of sea stars for blood gas and electrolyte analysis; was there any overlap between the animals analysed for blood gases and electrolytes?
Especially for the electrolytes for which only four replicates have been analysed, this gives a fairly high uncertainty for some of the values, exemplified by magnesium concentrations where three of the T0 values lie around 51 whereas the fourth (in the temperature group) is close to 65.
In line 164, the authors informed that sea stars were moved to individual 53 L tanks. I interpret this information as if each individual sea star had its own tank – or was it maybe each experimental group of five animals?
It is not quite clear to me how many water samples were analysed during the experiment.
Description of the experimental procedures:
The authors give an extremely detailed account of the experimental procedures. This section can definitely be shortened and still provide the necessary information. Some of the information could preferably be moved to some kind of Supplementary Information. This is also true for figure 2 and table 2.
Presentation and interpretation of data:
In table 4, the authors present mean values but no measure of variability. Standard deviations must be included. The table legend starts with the word ‘Changes’ - but the table shows the actual values at T0 and T1, not changes. Throughout the manuscript - especially in the tables - the number of digits presented must be adjusted to the deviation on the values. Including decimal points in values such as 486.4±50.7 does not make sense; should be given as 486±51.
It is obvious from table 4 that some of the T0 values vary more between the four groups than the variation brought about by the exposure to the various stressors between T0 and T1. Some of the changes attributed by the authors as statistically significant changes may therefore be the result of pure chance.
Statistical treatment:
The authors performed high number of paired t-tests. Did they consider the risk of obtaining false positives with a high number of tests.
Data themselves:
I don't understand why I cannot find identical T1 control values in table 4 and 5. I can for the blood gases, but not for the electrolytes. Why is that?
In table 6, what is the number of determinations of the electrolytes in the seawater? I simply cannot understand how the authors can manage to have almost 20% difference in e.g. sodium concentrations in the different groups (T0 mean: 394 mM; pH T1 mean: 337 mM).
Also in this table standard deviations should be given and the number of digits should be adjusted according to the standard deviation.
The authors should look across all of the values in their control groups to get a better impression of the variability and thereby obtain a more solid interpretation of the data
Author Response
The authors investigated the effect of changes in pH, oxygen concentration and temperature on coelomic parameters in the sea stars. The manuscript may end up being publishable after serious revision, but I have questions to the study design, some of the experimental procedures, the description of the experimental procedures, the presentation and interpretation of data, the statistical treatment and to the data themselves.
RESPONSE: THANK YOU FOR YOUR THOROUGH FEEDBACK, WE HAVE MADE THE CHANGES AS OUTLINED BELOW.
Study design and experimental procedure:
The authors take out samples from their experimental animals before ramp up of the parameters investigated and they ramp down after the T1 coelomic fluid sampling. The authors do not explain why they carry out this ramp down after the apparent finalisation of the experiment. Since they carried out this ramp down, it would have been very useful if they had taken out coelomic fluid samples after the ramp down to see if coelomic fluid parameters had returned to T0-values.
RESPONSE: WE CARRIED OUT THIS RAMP-DOWN OUT OF CONCERN FOR ANIMAL WELFARE, TO GIVE THE SEA STARS TIME TO ADJUST BACK TO THEIR BASELINE TANK CONDITIONS. TO CLARIFY THIS, WE HAVE ADDED “out of concern for animal welfare and to allow sea stars to adjust to their baseline housing parameters” TO LINES 231-232. BODY WALL BIOPSIES WERE TAKEN AFTER T1 COELOMIC FLUID SAMPLING, THIS WORK WILL BE PUBLISHED IN A SEPARATE MANUSCRIPT BUT THAT PRECLUDED EVALUATION OF COELOMIC FLUID FOLLOWING THE RAMP-DOWN PERIOD.
The authors choose to analyse coelomic parameters for a subset of the experimental animals, a procedure which I find very questionable. The authors argue that it has been done to minimise experimental costs but, in my experience, the factual costs of performing the analyses carried out are not very heavy for the individual analysis. The authors selected different numbers of sea stars for blood gas and electrolyte analysis; was there any overlap between the animals analysed for blood gases and electrolytes?
RESPONSE: THE ANALYSES COST NEARLY $50 PER ANIMAL AND UNFORTUNATELY, DID BECOME COST PROHIBITIVE. COELOMIC FLUID FROM ALL ANIMALS ANALYZED FOR ELECTROLYTES WERE ALSO ANALYZED FOR BLOOD GASES. THIS INFORMATION HAS BEEN ADDED TO LINES 326-327.
Especially for the electrolytes for which only four replicates have been analysed, this gives a fairly high uncertainty for some of the values, exemplified by magnesium concentrations where three of the T0 values lie around 51 whereas the fourth (in the temperature group) is close to 65.
RESPONSE: INDEED, THERE WAS HIGH VARIABILITY IN SOME VALUES. HOWEVER, THIS IS ACCURATELY REPORTED AND STILL USEFUL INFORMATION. WE WERE CAREFUL NOT TO OVERSTATE CONCLUSIONS FROM THE ELECTROLYTES, OSMOLALITY, AND CELL COUNTS DUE TO THE SMALL SAMPLE SIZE FOR THOSE ANALYSES.
In line 164, the authors informed that sea stars were moved to individual 53 L tanks. I interpret this information as if each individual sea star had its own tank – or was it maybe each experimental group of five animals?
RESPONSE: EACH SEA STAR HAD ITS OWN TANK FOR STATISTICAL INDEPENDENCE. THE WORD “individual” IS ON LINE 163. THAT WHOLE SENTENCE READS “Sea stars were randomly selected from group housing tanks and moved to individual 53 L tanks equipped with a heat exchanger, air stone, mechanical sock filter and fiber filtration media (Matala Water Technology, Taichung City, Taiwan) for biological filtration.”
It is not quite clear to me how many water samples were analysed during the experiment.
RESPONSE: IN TABLE 6, THE SAMPLE SIZE IS GIVEN IN THE TOP ROW. THE SEA STAR SAMPLES AND TANK WATER SAMPLES WERE PAIRED, THE N IS GIVEN FOR TANK WATER AND SEA STAR COELOMIC FLUID EACH. FOR EXAMPLE, N=14 IS REPORTED FOR T0 MEANING THERE WERE 14 TANK WATER SAMPLES AND 14 SEA STAR COELOMIC FLUID SAMPLES ANALYZED. WE HAVE CLARIFIED THIS IN THE TABLE 6 FIGURE LEGEND BY ADDING “Coelomic fluid and tank water samples were paired for analyses, the sample size (n) given represents the number of individuals for each type of analysis” TO LINES 483-484.
Description of the experimental procedures:
The authors give an extremely detailed account of the experimental procedures. This section can definitely be shortened and still provide the necessary information. Some of the information could preferably be moved to some kind of Supplementary Information. This is also true for figure 2 and table 2.
RESPONSE: THE EXPERIMENTAL EXPOSURE SECTION IS ONLY 38 LINES. THE INFORMATION PROVIDED IN THAT SECTION IS KEY TO THE REPEATABILITY OF THIS RESEARCH AND THUS, WE WOULD LIKE TO KEEP THAT LEVEL OF DETAIL IN THE BODY OF THE MANUSCRIPT RATHER THAN MOVE IT TO SUPPLEMENTAL MATERIAL. LIKEWISE, FIGURE 2 AND TABLE 2 SHOW INFORMATION THAT IS IMPORTANT TO THIS WORK AND SHOULD BE INCLUDED IN THE BODY OF THE MANUSCRIPT. FIGURE 2 SHOWS OBSERVABLE CHANGES TO SEA STAR APPEARANCE AND BEHAVIOR AND IS KEY TO UNDERSTANDING THE RESULTS PRESENTED. TABLE 2 SHOWS THE TANK TEMPERATURE VARIATION IN THE TEMPERATURE AND CONTROL GROUP TANKS.
Presentation and interpretation of data:
In table 4, the authors present mean values but no measure of variability. Standard deviations must be included. The table legend starts with the word ‘Changes’ - but the table shows the actual values at T0 and T1, not changes.
RESPONSE: STANDARD DEVIATION HAS BEEN ADDED TO TABLE 4. THE TABLE LEGEND HAS BEEN CHANGED TO “Mean ± standard deviation for…”
Throughout the manuscript - especially in the tables - the number of digits presented must be adjusted to the deviation on the values. Including decimal points in values such as 486.4±50.7 does not make sense; should be given as 486±51.
RESPONSE: THANK YOU FOR CATCHING THIS. NUMBERS WERE ADJUSTED BASED ON STANDARD LABORATORY REPORTING.
It is obvious from table 4 that some of the T0 values vary more between the four groups than the variation brought about by the exposure to the various stressors between T0 and T1. Some of the changes attributed by the authors as statistically significant changes may therefore be the result of pure chance.
RESPONSE: ALL OF THE T1 MEAN VALUES FALL OUTSIDE OF THE RANGE FOR T0 MEAN ± STANDARD DEVIATION. THIS IS CLEARER AFTER THE ADDITION OF STANDARD DEVIATION TO THE TABLE.
Statistical treatment:
The authors performed high number of paired t-tests. Did they consider the risk of obtaining false positives with a high number of tests.
RESPONSE: WE DID CONSIDER THE RISK OF FALSE POSITIVES WITH MULTIPLE COMPARISONS. A BONFERRONI CORRECTION REDUCES THE RISK OF TYPE I ERROR AT THE EXPENSE OF INCREASING TYPE II ERROR RATES AND IS NO LONGER RECOMMENDED (SEE ARMSTRONG 2014: https://onlinelibrary.wiley.com/doi/pdfdirect/10.1111/opo.12131?casa_token=bfW0eCHnnjUAAAAA:p783b_UnR7ZwHdMREWUkaZti6yhJCQpoc3OZ9P9ecI3IqiTpMR_NM5yGN3CGBt7PPDg5c1WIqFIJTQj5Fg). BECAUSE WE DID NOT MEET ANY OF THE CRITERIA FOR RECOMMENDING A BONFERRONI CORRECTION AND TO AVOID AN INCREASE IN TYPE II ERROR RATES, WE CHOSE TO NOT DO A CORRECTION FOR MULTIPLE COMPARISONS.
Data themselves:
I don't understand why I cannot find identical T1 control values in table 4 and 5. I can for the blood gases, but not for the electrolytes. Why is that?
RESPONSE: UNFORTUNATELY, SOME T0 SAMPLES FOR ELECTROLYTES, OSMOLALITY AND CELL COUNTS HAD TO BE EXCLUDED FROM ANALYSES AFTER SAMPLE QUALITY WAS AFFECTED DUE TO LOGISTICAL ISSUES. SINCE TABLE 4 IS A COMPARISON BETWEEN T0 AND T1, ONLY SEA STARS THAT HAD RESULTS FROM BOTH TIME POINTS WERE INCLUDED. THEREFORE, THERE WERE FOUR SEA STARS PER GROUP FOR TABLE 4 AND SEVEN SEA STARS PER GROUP IN TABLE 5.
In table 6, what is the number of determinations of the electrolytes in the seawater? I simply cannot understand how the authors can manage to have almost 20% difference in e.g. sodium concentrations in the different groups (T0 mean: 394 mM; pH T1 mean: 337 mM).
Also in this table standard deviations should be given and the number of digits should be adjusted according to the standard deviation.
RESPONSE: THE NUMBER OF TANK WATER SAMPLES TESTED IS IN THE FIRST ROW OF TABLE 6. THE FOLLOWING SENTENCES ADDRESSING THE HIGH VARIABILITY OF ELECTROLYTES HAVE BEEN ADDED TO THE DISCUSSION (LINES 654-658): “The high variability of some analytes, as evidenced by the high standard deviation (e.g. magnesium and sodium), may have resulted in statistical differences due to chance rather than being representative as a response to environmental stressors. However, all statistically significant T1 mean values fell outside of the range for T0 mean ± standard deviation, thus supporting the observed statistical differences.” TO ADDRESS YOUR CONCERN ABOUT VARIATION IN THE ELECTROLYTES, WE HAVE INCLUDED A SUPPLEMENTARY TABLE (TABLE S1) SHOWING THE COEFFICIENT OF VARIATION FOR THESE ANALYSES RANGED FROM 2.0-2.4%. WE HAVE ADDED ADDITIONAL INFORMATION ABOUT CALIBRATION (LINES 222-225 AND 258-260).
The authors should look across all of the values in their control groups to get a better impression of the variability and thereby obtain a more solid interpretation of the data
RESPONSE: A SENTENCE ADDRESSING THE HIGH VARIABILITY OF ELECTROLYTES HAS BEEN ADDED TO THE DISCUSSION (LINES 654-658). WE FEEL THAT THE INTERPRETATION OF THE DATA HAS BEEN PRESENTED CAUTIOUSLY TO AVOID OVERINTERPRETATION.